# On the Expressive Power of Tree-Structured Probabilistic Circuits

**Lang Yin**
Department of Computer Science
University of Illinois Urbana-Champaign
`langyin2@illinois.edu`

**Han Zhao**
Department of Computer Science
University of Illinois Urbana-Champaign
`hanzhao@illinois.edu`

## Abstract

Probabilistic circuits (PCs) have emerged as a powerful framework to compactly represent probability distributions for efficient and exact probabilistic inference. It has been shown that PCs with a general directed acyclic graph (DAG) structure can be understood as a mixture of exponentially (in its height) many components, each of which is a product distribution over univariate marginals. However, existing structure learning algorithms for PCs often generate tree-structured circuits or use tree-structured circuits as intermediate steps to compress them into DAG-structured circuits. This leads to the intriguing question of whether there exists an exponential gap between DAGs and trees for the PC structure. In this paper, we provide a negative answer to this conjecture by proving that, for $n$ variables, there exists a quasi-polynomial upper bound $n^{O(\log n)}$ on the size of an equivalent tree computing the same probability distribution. On the other hand, we also show that given a depth restriction on the tree, there is a super-polynomial separation between tree and DAG-structured PCs. Our work takes an important step towards understanding the expressive power of tree-structured PCs, and our techniques may be of independent interest in the study of structure learning algorithms for PCs.

## 1 Introduction

Probabilistic circuits (PCs) [7, 29], also commonly known as sum product networks (SPNs) [24], are a type of deep graphical model that allow exact probabilistic inference efficiently in linear time with respect to the size of the circuit. Like other deep models, the parameters of a PC can be learned from data samples [38]. Because of these desirable properties, they have been increasingly applied in various contexts, including generative modeling [35], image processing [2, 34], robotics [30], planning [25] and sequential data including both textual and audio signals [6, 23]. Compared with deep neural networks, the sum and product nodes in PCs admit clear probabilistic interpretation [36] of marginalization and context-specific statistical independence [3], which opens the venue of designing efficient parameter learning algorithms for PCs, including the expectation-maximization (EM) algorithm [12], the convex-concave procedure (CCCP) [38], and the variational EM algorithm [37].

Perhaps one of the most important properties of PCs is that they can be understood as a mixture of exponentially (in its height) many components, each of which is a product distribution over univariate marginals [38]. Intuitively, each sum node in PC can be viewed as a hidden variable that encodes a mixture model [36] and thus a hierarchy of sum nodes corresponds to an exponential number of components. This probabilistic interpretation of PCs has led to a number of interesting structure learning algorithms [1, 9, 13, 17, 22, 27]. However, almost all of the existing structure learning algorithms for PCs output tree-structured circuits, or use tree-structured circuits as intermediates to compress them into DAG-structured circuits. Because of the restricted structure of trees, such algorithms do not fully exploit the expressive power of PCs with general DAG structures, and often

38th Conference on Neural Information Processing Systems (NeurIPS 2024).

output tree-structured PCs with exceedingly large sizes [13]. Yet, from a theoretical perspective, it remains open whether there truly exists an exponential gap between DAGs and trees for the PC structure. Being able to answer this question is important for understanding the expressive power of tree-structured PCs, and may also lead to new insights in structure learning algorithms for PCs.

## 1.1 Our Contributions

In this work we attempt to answer the question above by leveraging recent results in complexity theory [11, 26, 33]. Our contributions are two-folds: an upper and lower bound for the gap between tree and DAG-structured PCs. In what follows we will first briefly state our main results and then introduce the necessary concepts and tools to tackle this problem.

**An Upper Bound** In Section 3, inspired by earlier works in Valiant et al. [33] and Raz and Yehudayoff [26], we show that, for a network polynomial that can be computed efficiently with a DAG-structured PC, there always exists a tree-structured PC of a quasi-polynomial size to represent it. An informal version of our main result for this part is stated below.

**Theorem 1.1** (Informal). *Given a network polynomial of $n$ variables, if this polynomial can be computed efficiently by a PC of size $\mathrm{poly}(n)$, then there exists an equivalent tree-structured PC of depth $O(\log n)$ and of size $n^{O(\log n)}$ that computes the same network polynomial.*

We prove this result by adapting the proof in Raz and Yehudayoff [26]. Our construction involves two phases: Phase one applies the notion of *partial derivatives* for general arithmetic circuits to represent intermediate network polynomials alternatively, and construct another DAG-structured PC using those alternative representations; we will provide fine-grained analysis on the new DAG, such that its depth is $O(\log n)$ and its size is still $\mathrm{poly}(n)$. Phase two applies the standard duplicating strategy for all nodes with more than one parent to convert the new DAG into a tree. This strategy will lead to an exponential blowup for an arbitrary DAG with depth $D$, since the size of the constructed tree will be $n^{O(D)}$. However, note that the DAG constructed in the first phase has depth $O(\log n)$. Combining it with the duplicating strategy, we will be able to construct an equivalent tree-structured PC of size upper bounded by $n^{O(\log n)}$, as desired.

The original algorithm in Raz and Yehudayoff [26] only reduces the depth to $O(\log^2 n)$ due to their restriction on the graph of using nodes with at most two children. This restriction is not necessary for PCs, and by avoiding it, we show that the depth can be further reduced to $O(\log n)$ with a slight modification of the original proof in Raz and Yehudayoff [26].

**A Lower Bound** In Section 4, we show that under a restriction on the depth of the trees, there exists a network polynomial that can be computed efficiently with a DAG-structured PC, but if a tree-structured PC computes it, then the tree must have a super-polynomial size. The following informal theorem states our main result for this part, which will be formally addressed in Section 4.

**Theorem 1.2** (Informal). *Given $n$ random variables, there exists a network polynomial on those $n$ variables, such that it can be efficiently computed by a PC of size $O(n \log n)$ and depth $O(\log n)$, but any tree-structured PC with depth $o(\log n)$ computing this polynomial must have size at least $n^{\omega(1)}$.*

Our result is obtained by finding a reduction to Fournier et al. [11]. We first fix an integer $k$ and a network polynomial of degree $n = 2^{2k}$. To show that the polynomial is not intrinsically difficult to represent, i.e., the minimum circuit representing it shall be efficient, we explicitly construct a PC of depth $O(\log n)$ and size $O(n \log n)$. Next, suppose via a black box, we have a minimum tree-structured PC of depth $o(\log n)$ computing this polynomial. After removing some leaves from that minimum tree but maintaining its depth, we recover a regular arithmetic tree that computes a network polynomial of degree $\sqrt{n} = 2^k$. Moreover, as shown in [11], if an arithmetic tree with depth $o(\log n)$ computes this low-degree polynomial, then the size of the tree must be at least $n^{\omega(1)}$. Our operations on the minimum tree PC must reduce its size; therefore, the original tree must have a larger size than $n^{\omega(1)}$, and this fact concludes our proof.

## 1.2 More Related Work

There is an extensive literature on expressive efficiency of network structures for PCs and probabilistic generating circuits (PGCs) [35], another probabilistic graphic model. Very recently, it was shown in

Broadrick et al. [4] that PCs with negative weights are as expressive as PGCs. The investigation on PCs has started as early as in Delalleau and Bengio [8] and later in Martens and Medabalimi [19]. In neural networks and variants, this topic, along with the relationship between expressive efficiency and depth/width, has attracted many interests as well [10, 16, 18, 20, 21, 28, 31, 32]. In particular, Martens and Medabalimi [19, Theorem 34] has shown that there exists a network polynomial with a super-polynomial minimum tree expression, but it is unclear whether the same polynomial can be computed by a polynomial-sized DAG. Our work provides a positive answer to this question. For arbitrary network polynomials, finding a minimum DAG-structured PC is reducible to a special case of the *minimum circuit size problem* for arithmetic circuits, which remains to be a longstanding open problem in circuit complexity.

## 2  Preliminaries

We first introduce the setup of probabilistic circuits and relevant notation used in this paper.

**Notation**  A rooted directed acyclic (DAG) graph consists a set of nodes and directed edges. For such an edge $u \to v$, we say that $u$ is a *parent* of $v$, and $v$ is a *child* of $u$. We use $\mathrm{Ch}(u)$ to denote the set of children of the node $u$. We say there is a directed path from $a$ to $b$ if there is an edge $a \to b$ or there are nodes $u_1, \cdots, u_k$ and edges $a \to u_1 \to \cdots \to u_k \to b$; in this case, we say that $a$ is an *ancestor* of $b$ and that $b$ is a *descendant* of $a$. If two vertices $v$ and $w$ are connected via a directed path, we call the number of edges in a shortest path between them as the *distance* between them, denoted by $\mathrm{dist}(v, w)$. A directed graph is *rooted* if one and only one of its nodes has no incoming edges. A *leaf* in a DAG is a node without outgoing edges. A *cycle* in a directed graph is a directed path from a node to itself, and a directed graph without directed cycles is a DAG. For two disjoint sets $A$ and $B$, we will denote their disjoint union by $A \sqcup B$ to emphasize their disjointedness.

Clearly, each directed graph has an underlying undirected graph obtained by removing arrows on all edges. Although by definition, a DAG cannot have a directed cycle, but its underlying undirected graph may have an undirected cycle. If the underlying undirected graph of a DAG is also acyclic, then that DAG is called a **directed tree**. Every node in a directed tree has at most one parent. If two nodes share a parent, one is said to be a *sibling* of the other.

**Complexity Classes**  In what follows we introduce the necessary complexity classes that will be used throughout the paper. Let $f(n)$ be the runtime of an algorithm with input size $n$.

- A function $f(n)$ is in the **polynomial** class $\mathrm{poly}(n)$ if $f(n) \in O(n^k)$ for a constant $k \in \mathbb{N}$.
- A function $f(n)$ is in the **super-polynomial** class if $f(n)$ is not asymptotically bounded above by any polynomial. Formally, this requires $f(n) \in \omega(n^c)$ for any constant $c > 0$, i.e. $\lim_{n \to \infty} \frac{f(n)}{n^c} = \infty$ for any $c > 0$.
- A function $f(n)$ is in the **quasi-polynomial** class if it can be expressed in the form $2^{\mathrm{poly}(\log n)}$.
- A function $f(n)$ is in the **exponential** class if it can be expressed in the form $2^{O(\mathrm{poly}(n))}$.

**Probabilistic Circuits**  A probabilistic circuit (PC) is a probabilistic model based on a rooted DAG. Without loss of generality, in our work, we focus on PCs over Boolean random variables. We first introduce the notion of *network polynomial*. For each Boolean variable $X$, we use the corresponding lower case alphabet $x$ to denote the indicator of $X$, which is either 0 or 1; for the same variable, $\bar{x}$ represents the negation. In many cases, we use $1 : N$ to denote the index set $[N]$. A PC over Boolean variables $\{X_1, \cdots, X_n\}$ computes a polynomial over the set of indicators $\{x_1, \cdots x_n, \bar{x}_1, \cdots, \bar{x}_n\}$; we will refer this polynomial as the *network polynomial*. In the network, the leaves are indicators of variables, and all other nodes are either sum or product nodes; a node that is not a leaf may also be called an *internal* node. Each internal node computes a polynomial already: a sum node computes a weighted sum of the polynomials computed by its children, and a product node computes the product of the polynomials computed by its children. A PC is said to be *normalized* if the weights of the outgoing edges from a sum node sum to one. It was shown in Zhao et al. [36] that, every unnormalized PC can be transformed into an equivalent normalized PC within linear time.

To represent valid probability distributions, a PC must satisfy two structural properties: *decomposability* and *smoothness*. To define them, we need to define the *scope* of a node, which is the set of

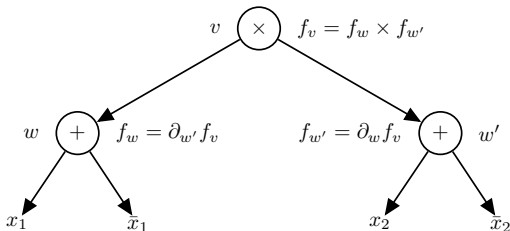

Figure 1: Partial derivatives of sum nodes.

variables whose indicators are descendants of that node. For a node $v$, if the indicator $x$ of the variable $X$ is one of descendants of $v$, then $X \in \text{scope}(v)$; more generally, $\text{scope}(v) = \cup_{v' \in \text{Ch}(v)} \text{scope}(v')$.

**Definition 2.1** (Decomposability and Smoothness). *A PC is decomposable if and only if for every product node $v$ and any pair of its children $v_1$ and $v_2$, we have $\text{scope}(v_1) \cap \text{scope}(v_2) = \emptyset$. A PC is smooth if and only if for each sum node, all of its children have the same scope.*

In this paper we restrict our attention to PCs that are both decomposable and smooth, since otherwise we can always transform a PC into an equivalent one that is both decomposable and smooth in quadratic time [36]. The *degree* of a monomial is the sum of the exponents of all its variables, and the *degree* of a polynomial $f$, denoted by $\deg(f)$ is the highest degree among its constituent monomials. A polynomial is said to be *homogeneous* if all of its monomials have the same degree. A PC is said to be *homogeneous* if all of its sum and product nodes compute a homogeneous polynomial. Later, we will show that decomposability and smoothness imply homogeneity, and vice versa with mild conditions. For a node $v$ in a PC, we use $\deg(v)$ to denote $\deg(f_v)$. As emphasized earlier, this paper investigates the quantitative relationship between a DAG and a tree, which are both PCs and represent the same probability distribution. To make the terminology uncluttered, we will call the former a *DAG-structured PC*, and the latter a *tree PC*. If a tree PC computes the same network polynomial as a DAG-structured PC, then the tree PC is said to be an *equivalent tree PC* with respect to that DAG-structured PC.

Unless specified otherwise, we will use $\Phi$ to denote the entire PC in consideration and $f$ the network polynomial computed by the root. For each node $v$, the sub-network rooted at $v$ is denoted by $\Phi_v$ and the polynomial computed by $v$ is $f_v$. The set of variables in $f_v$ is $X_v$, which is a subset of $\{X_1, \cdots, X_n\}$. The size of the network $\Phi$, denoted by $|\Phi|$, is the number of nodes and edges in the network. The depth of $\Phi$, denoted by $D(\Phi)$, is its maximum length of a directed path.

**Partial Derivatives** In the process of proving the upper bound, a key notion named *partial derivative* is frequently used and formally defined below.

**Definition 2.2** (Partial Derivative). *For two nodes $v$ and $w$ in a network $\Phi$, the partial derivative of the polynomial $f_v$ with respect to the node $w$, denoted by $\partial_w f_v$, is constructed by the following steps:*

1. *Substitute the polynomial $f_w$ by a new variable $y$.*
2. *Compute the polynomial computed by $v$ with the variable $y$; denote the new polynomial by $\bar{f}_v$. Due to decomposability, $\bar{f}_v$ is linear in $y$.*
3. *Define the partial derivative $\partial_w f_v = \frac{\partial \bar{f}_v}{\partial y}$.*

Observe that the chain rule in calculus also holds for our notion here, and therefore leads to the two following facts.

- Let $v$ be a sum node with children $v_1$ and $v_2$, and the edges have weight $a_1$ and $a_2$, respectively, then by definition $f_v = a_1 f_{v_1} + a_2 f_{v_2}$. For any other node $w$, the partial derivative is $\partial_w f_v = a_1 \cdot \partial_w f_{v_1} + a_2 \cdot \partial_w f_{v_2}$.

- Similarly, let $v$ be a product node with children $v_1$ and $v_2$, then $f_v = f_{v_1} \cdot f_{v_2}$. For any other node $w$, we have $\partial_w f_v = f_{v_1} \cdot \partial_w f_{v_2} + f_{v_2} \cdot \partial_w f_{v_1}$.

The partial derivative has been proven to be a powerful tool in the field of complexity theory and combinatorial geometry [14, 15]. Readers are welcome to refer Chen et al. [5] for more details and extensive background. An illustration is provided in Figure 1.

**Arithmetic Circuits**   An arithmetic circuit, aka algebraic circuit, is a generalization of a probabilistic circuit. Such a circuit shares the same structure as a PC and also computes a network polynomial. If an arithmetic/algebraic circuit is a directed tree, then we call it an *arithmetic/algebraic formula*. In the proof of the lower bound, the notion of *monotonicity* of a formula is essential, whose definition relies on the concept *parse tree*.

**Definition 2.3** (Parse Tree). *A parse tree of a formula $\Phi$ is a sub-formula of $\Phi$ which corresponds to a monomial of $f$, the network polynomial computed by $\Phi$. Parse trees of $\Phi$ is defined inductively by the following process:*

- *If the root of $\Phi$ is a sum node, a parse tree of $\Phi$ is obtained by taking a parse tree of **one of its children** together with the edge between the root and that child.*
- *If the root of $\Phi$ is a product node, a parse tree of $\Phi$ is obtained by taking a parse tree of **each of its children** together with **all** outgoing edges from the root.*
- *The only parse tree of a leaf is itself.*

**Definition 2.4** (Monotonicity). *An algebraic formula is monotone if the monomial computed by any of its parse trees has a non-negative coefficient in the network polynomial.*

## 3   A Universal Upper Bound

In this section, we present our first main result, which provides a universal upper bound on the size of an equivalent tree versus a DAG-structured PC.

**Theorem 3.1.** *For any given DAG-structured PC over $n$ variables and of size $\mathrm{poly}(n)$, there exists an equivalent tree-structured PC of size $n^{O(\log n)}$ nodes and of depth $O(\log n)$, computing the same polynomial.*

As discussed earlier, our constructive proof heavily relies on deeper properties of partial derivatives, and applying them to represent sub-network polynomials. Our strategy, inspired by Raz and Yehudayoff [26], will be efficient if the circuit being considered is a binary circuit, i.e. every node has at most two children. While such structure is rare for natural networks, we make the following observation, that an arbitrary PC can always be transformed to a binary one with a polynomial increase in size and depth. The proof and an illustrating figure will appear in Appendix A.

**Lemma 3.2.** *Given a DAG-structured PC $\Phi$, we may transform it into another DAG $\Phi'$ that computes the same network polynomial and every node in $\Phi'$ has at most two children. Moreover, the differences between the sizes and depths of $\Phi'$ and $\Phi$ are only in polynomial size.*

Therefore, for the remaining discussion in this section, we will assume without loss of generality, that a given PC is binary. During the conversion process towards a binary circuit, some *intermediate* nodes may be created to ensure no sum node is connected to another sum node and no product node is connected to another product node. The set of those intermediate nodes is denoted by $\Phi_1$, and will be present in our later discussions. Next, we present a key property of partial derivatives, which holds for any (including non-binary) PC.

**Lemma 3.3.** *Given a PC $\Phi$, if $v$ and $w$ are two nodes in $\Phi$ such that $\partial_w f_v \neq 0$, then $\partial_w f_v$ is a homogeneous polynomial over the set of variables $X_v \setminus X_w$ of degree $\deg(v) - \deg(w)$.*

The next lemma tells that, given a product node, its partial derivative with another node with a restriction on degree can be expressed using its children.

**Lemma 3.4.** *Let $v$ be a product node and $w$ be any other node in a PC $\Phi$, and $\deg(v) < 2\deg(w)$. The children of $v$ are $v_1$ and $v_2$ such that $\deg(v_1) \geq \deg(v_2)$. Then $\partial_w f_v = f_{v_2} \cdot \partial_w f_{v_1}$.*

To construct the quasi-polynomial tree, the key is to compress many nodes with partial derivatives. Fundamentally, we will use the following results to show that such compression works because each node, and each partial derivative of any node with any other, can be more concisely represented using partial derivatives. The key question is to find eligible nodes, so that taking partial derivatives with respect to them will lead to compact expressions. Inspired by the observation in Raz and Yehudayoff [26], we define the following set $\mathbf{G}_m$, which will be critical in our later process.

---

**Algorithm 1:** Construction of $\Psi$

---

**Data:** The original DAG-structured PC $\Phi$ with $n$ variables of size $\text{poly}(n)$, and the set of its nodes $\mathcal{V}$

**Result:** Another DAG-structured PC $\Psi$ of size $\text{poly}(n)$ and depth $O(\log n)$

$i \leftarrow 0; \mathcal{V} \leftarrow \emptyset; \mathcal{P} \leftarrow \emptyset$.

**for** $i = 0, 1, \lceil \log n \rceil - 1$ **do**

    Fix $m_1 \leftarrow 2^i$;

    Find all nodes $v$ such that $2^i < \deg(v) \leq 2^{i+1}$, and place them in $\mathcal{V}$;

    Find all pairs of nodes $(u, w)$ such that $2^i < \deg(u) - \deg(w) \leq 2^{i+1}$ and $\deg(u) < 2\deg(w)$, and place them in $\mathcal{P}$;

    **for** *every* $v \in \mathcal{V}$ **do**

        Find all nodes in $\mathbf{G}_{m_1}$ and compute $f_v$ using Equation 15;

    **end**

    **for** *every pair of nodes* $(u, w) \in \mathcal{P}$ **do**

        Fix $m_{2,(u,w)} \leftarrow 2^i + \deg(w)$;

        Find all nodes in $\mathbf{G}_{m_{2,(u,w)}}$ and compute $\partial_w f_v$ using Equation 17;

    **end**

    $\mathcal{V} \leftarrow \emptyset; \mathcal{P} \leftarrow \emptyset$.

**end**

---

**Definition 3.5.** *Given a PC $\Phi$ and an integer $m \in \mathbb{N}$, the set $\mathbf{G}_m$ is the collection of product nodes $t$ in $\Phi$ with children $t_1$ and $t_2$ such that $m < \deg(t)$ and $\max\{\deg(t_1), \deg(t_2)\} \leq m$.*

With this set, we may choose a set of nodes as variables for partial derivatives for any node in a PC, and the following two lemmas respectively illustrate: 1) the compact expression of the sub-network polynomial $f_v$ for any node $v$ in a PC; 2) the compact expression of $\partial_w f_v$ given two nodes $v$ and $w$ with a degree restriction. It is easy to observe that $\Phi_1 \cap \mathbf{G}_m = \emptyset$.

We now present two key lemmas that will be central to the proof for the upper bound. Specifically, they spell out the alternative representations for the network polynomial of any node, and the partial derivative of any pair of nodes.

**Lemma 3.6** ([26]). *Let $m \in \mathbb{N}$ and a node $v$ such that $m < \deg(v) \leq 2m$, then $f_v = \sum_{t \in \mathbf{G}_m} f_t \cdot \partial_t f_v$.*

**Lemma 3.7** ([26]). *Let $m \in \mathbb{N}$, and $v$ and $w$ be two nodes such that $\deg(w) \leq m < \deg(v) < 2\deg(w)$, then $\partial_w f_v = \sum_{t \in \mathbf{G}_m} \partial_w f_t \cdot \partial_t f_v$.*

### 3.1 Construction of $\Psi$, another DAG-structured PC with restriction on depth

Given a binary DAG-structured PC $\Phi$ with $n$ variables and $\text{poly}(n)$ nodes, we explicitly construct a tree PC with size $n^{O(\log n)}$ and depth $O(\log n)$. Specifically, the construction takes two main steps:

1. Transform $\Phi$ to another DAG-structured PC $\Psi$ with size $\text{poly}(n)$ and depth $O(\log n)$.
2. Apply a simple duplicating strategy to further convert $\Psi$ to a tree with size $n^{O(\log n)}$ and the same depth of $\Psi$.

We will later show that step two can be simply done. Step one, however, needs much more careful operations. Each iteration, starting from $i = 0$, again needs two steps:

1. Compute $f_v$ for each node $v$ such that $2^{i-1} < \deg(v) \leq 2^i$ using the compact expression illustrated earlier. We will show that, computing one such polynomial adds $\text{poly}(n)$ nodes and increases the depth by at most two on $\Psi$. This new node representing $f_v$ will be a node in $\Psi$, denoted by $v'$.
2. Compute all partial derivatives $\partial_w f_u$ for two non-variable nodes $u$ and $w$ in $\Phi$, such that $u$ is an ancestor of $w$ and $2^{i-1} < \deg(u) - \deg(w) \leq 2^i$ and $\deg(u) < 2\deg(w)$. Like those new nodes representing sub-network polynomials from $\Phi$, this new node representing a partial derivative will also be a node in $\Psi$, denoted by $(u, w)$. We will show that computing a partial derivative with respect to each pair adds $\text{poly}(n)$ nodes and increases the depth by at most two on $\Psi$.

**Algorithm 2:** Transforming a rooted DAG to a tree

---

**Data:** A rooted DAG of size $S$ and depth $D$, and the set of its nodes $\mathcal{V}$
**Result:** A tree of size $O(S^D)$ and depth $D$
**for** *every node $v$ in $\mathcal{V}$* **do**
    **if** $\mathrm{InDeg}(v) > 1$ **then**
        Duplicate the tree rooted at $v$ for $\mathrm{InDeg}(v) - 1$ times;
        Construct an outgoing edge from each parent of $v$ to itself.
    **end**
**end**

---

The process is summarized in Algorithm 1. Before presenting the construction, we first confirm the quantitative information of $\Psi$, the output of the algorithm. The first observation is the number of iterations: The degree of the root of $\Phi$ is $n$, so at most $\log n$ iterations are needed for the entire process. Each iteration only increases the size of the updated circuit by $\mathrm{poly}(n)$ and the depth by a constant number. Consequently, the final form of $\Psi$ has size $\mathrm{poly}(n)$ and depth $O(\log n)$.

We now provide an inductive construction of $\Psi$ starting from $i = 0$. After each step, it is necessary to verify the validity of the updated $\Psi$. Although decomposability is easy to verify, smoothness is less straightforward. To tackle this, we argue that the final state of $\Psi$ is homogeneous, i.e. every node in $\Psi$ computes a homogeneous polynomial, and consequently $\Psi$ is smooth due to the following lemma.

**Lemma 3.8.** *If a decomposable PC contains $n$ variables and computes a polynomial of degree $n$, then it is homogeneous if and only if it is smooth.*

**Iteration zero** $(i = 0)$**:** During this iteration, for the first step, we only need to consider nodes $v$ such that $0.5 < \deg(v) \leq 1$; the degree of any node must be an integer, so we must have $\deg(v) = 1$, i.e. $v$ represents an affine polynomial. Without loss of generality, we may assume all such affine nodes are sum nodes with strictly more than one child. Indeed, if a product node represents an affine polynomial, then it must only have exactly one child, which must be a leaf node; in this case, we may remove this product node and connect that leaf to the parents of the original product node. Similarly, if a sum node represents an affine polynomial and has exactly one child, then that child must also be a leaf node, hence we may again remove the sum node and connect that leaf to the parents of the original sum node. Due to smoothness, such an affine node $v$ must represent a polynomial in the form $ax + (1 - a)\bar{x}$, where $x$ is the indicator of a variable, and $0 < a < 1$. Therefore, the depth of each sub-network $\Phi_v$ is only one. By duplicating all such affine nodes onto $\Psi$, we add at most $\mathrm{poly}(n)$ nodes and increase the depth by one only.

Next, for step two, we only need to consider pairs of nodes $(u, w)$ such that $\deg(u) - \deg(w) \leq 1$. Thanks to Lemma 3.3, we know that $\partial_w f_u$ is affine. For each pair satisfying the restriction, we create a sum node $(u, w)$ whose sub-network $\Phi_{(u,w)}$ has size three and depth one. By moving all such sub-networks to $\Psi$ for each eligible pair, we again add at most $\mathrm{poly}(n)$ nodes and increase the depth by one to $\Psi$.

**Iteration $i + 1$:** Suppose, after all previous iterations, we have already computed all sub-network polynomials $f_v$ for nodes $v$ such that $\deg(v) \leq 2^i$, and all partial derivatives $\partial_w f_u$ for pairs of nodes $(u, w)$ such that $\deg(u) - \deg(w) \leq 2^i$ and $\deg(u) \leq 2\deg(w)$. Like the base case, step $i + 1$ takes two steps: The first step computes $f_v$ for eligible nodes, and the second step concerns partial derivatives for eligible pairs of nodes. Because the analysis of the two steps during this iteration is highly involved, we will discussion the construction in details in Appendix A.7.

## 3.2 Construction of the Quasi-polynomial Tree

We conclude the proof of Theorem 3.1 in this section by transforming the newly constructed $\Psi$ into a quasi-polynomial tree. The transformation is a simple application of the *naive duplication* strategy, which will be illustrated below. In summary, given a $\mathrm{poly}(n)$-sized DAG, the size of the transformed tree directly depends on the depth of the original DAG. The process of the duplication is briefly summarized in Algorithm 2, and the detailed process of the entire transformation from the original $\Phi$ to the final tree is described in Algorithm 5.

**Duplication Strategy**   Given a DAG-structured PC of size $V$ and depth $D$, a natural algorithm to a tree is that, if a node $v$ has $k > 1$ parents, then duplicate the sub-tree rooted at $v$ for $k - 1$ times, and connect each duplicated sub-tree to a parent of $v$. Indeed this algorithm generates a tree computing the same function as the original DAG does, but in the worst case we have to duplicate the entire graph $O(V)$ times and such iterative duplication may be executed for every layer from the first to layer $D$. Therefore, in the worst case, the final tree has size $O(V^D)$.

The construction of $\Psi$ shows that its size is $O(n^3)$ and depth is $O(\log n)$. Using the naive duplication, we obtain that the size of the final tree is $n^{O(\log n)}$.

## 4   A Conditional Lower Bound

In this section, we present our second main result, which provides a lower bound on the tree complexity of a network polynomial given a restriction on the depth of the tree. Obtaining a lower bound for the problem of circuit complexity is in general a more difficult problem than obtaining an upper bound because one cannot achieve this goal by showing the failure of a single algorithm. Instead, one must construct a specific polynomial, and confirm that no algorithm can produce an equivalent tree of size lower than the desired lower bound. However, thanks to some recent results in circuit complexity theory, such a separation is ensured if the tree PC has a bounded depth. The main result in this section is presented below, stating that, there is a concrete network polynomial that cannot be represented by a polynomial-sized tree-structured PC if the depth of the tree is restricted.

**Theorem 4.1.** *Given an integer $k \geq 1$ and $n = 2^{2k}$, there exists a network polynomial $P \in \mathbb{R}[x_1, \cdots, x_n, \bar{x}_1, \cdots, \bar{x}_n]$ of degree $n = 2^{2k}$, such that any probabilistic tree of depth $o(\log n) = o(k)$ computing $P$ must have size $n^{\omega(1)}$.*

Note that if the polynomial $P$ is innately difficult to be represented by PCs, i.e., if it cannot even be represented efficiently by DAG-structured PCs, then separation is not shown. To show separation, $P$ should be efficiently computed by a DAG-structured PC, but any tree-structured PC representing $P$ must have a strictly larger size. Our next construction, described with more details in Algorithm 3, confirms a separation by constructing an efficient DAG-structured PC $P^*$ that computes $P$. This PC has size $O(n \log n)$ and depth $2k = 2 \log n$, where $k$ is the integer given in Theorem 4.1. The next proposition confirms the validity of $P^*$, and the proof is in Appendix B.

**Proposition 4.2.** *The tree PC $P^*$ is decomposable and smooth.*

It is easy to check that $P^*$ has the correct size and depth as described earlier. Before adding leaf nodes, the algorithm in total constructs $\sum_{r=0}^{2k} 2^r = 2^{2k+1} - 1 = 2n - 1$ nodes. Finally, observe that during the construction of leaf nodes, each negation indicator is added exactly $k$ times: At a layer containing only product nodes, if a negation indicator is added to a product node $v$ at this layer, then it will next be added to the sibling of the grandparent of $v$. Because each product node has exactly one sibling, the negation indicator for a random variable is duplicated exactly $k$ times, and finally the total size is $2n - 1 + kn = O(kn) = O(n \log n)$. The depth $O(k)$ is also apparent from the algorithm. We therefore conclude that $P$ can be efficiently computed by a polynomial sized tree PC for an unrestricted depth.

However, the efficiency would be less optimal if we restrict the depth to $o(k)$. To show this, we design a reduction from our problem for PCs to a well-studied problem on arithmetic circuits. Our proof essentially argues that, for any minimum-sized tree-structured PC that computes $P$, we can obtain its sub-tree that computes a polynomial, and that polynomial has been recently proven to not be able to be represented by a polynomial-sized tree-structured PC. This recent result is stated below.

**Theorem 4.3** ([11]). *Let $n$ and $d = d(n)$ be growing parameters such that $d(n) \leq \sqrt{n}$. Then there is a monotone algebraic formula $F$ of size at most $n$ and depth $O(\log d)$ computing a polynomial $Q \in \mathbb{F}[x_1, \cdots, x_n]$ of degree at most $d$ such that any monotone formula $F'$ of depth $o(\log d)$ computing $Q$ must have size $n^{\omega(1)}$.*

The proof of the lower bound for PCs in Theorem 4.1 is to show that, for any $\Pi$, a minimum tree-structured PC with depth $o(k)$ that computes $P$, the polynomial in the statement of Theorem 4.1, we can always obtain a smaller-sized arithmetic formula $\Pi'$ with the same depth that computes the polynomial $Q$ in the statement of Theorem 4.3. The size of $\Pi'$ is super-polynomial due to

---

**Algorithm 3:** Construction of $P^*$, an efficient PC for $P$ without a depth constraint

---

**Data:** A positive integer $k$, the number $2^{2k}$, a set $\{x_1, \cdots, x_{2^{2k}}, \bar{x}_1, \cdots, \bar{x}_{2^{2k}}\}$ of $2^{2k+1}$ indicators

**Result:** A tree PC of size $O(n \log n)$ and depth $2k = 2 \log n$

$j \leftarrow 0$

Place all non-negation indicators $x_1, \cdots, x_{2^{2k}}$ at the bottom layer;

Label them as $L_{0,1}, \cdots, L_{0,2^{2k}}$.

**for** $i = 1, \cdots, 2 \log n$ **do**

    **if** $i$ *is odd* **then**

        **while** $j < 2^{2k-i}$ **do**

            Construct a product node labelled by $L_{i,(j/2)}$ and two outgoing edges from the new node to $L_{i-1,j-1}$ and $L_{i-1,j}$;

            $j \leftarrow j + 2$;

        **end**

        **for** *every odd integer* $q = 1, 3, \cdots, 2^{2k-i} - 1$ **do**

            Add the leaves representing negation indicators $\{x_z\}$ for all $z \in \mathrm{scope}(L_{i,q+1})$ as children of $L_{i,q}$;

            Add the leaves representing negation indicators $\{x_z\}$ for all $z \in \mathrm{scope}(L_{i,q})$ as children of $L_{i,q+1}$;

        **end**

    **end**

    **if** $i$ *is even* **then**

        **while** $j < 2^{2k-i}$ **do**

            Construct a sum node labelled by $L_{i,(j/2)}$ and two outgoing edges from the new node to $L_{i-1,j-1}$ and $L_{i-1,j}$;

            $j \leftarrow j + 2$;

        **end**

    **end**

**end**

---

Theorem 4.3, and as a result, the size of $\Pi$ cannot be smaller. In other words, our proof involves a reduction from the PC problem to the AC problem. Before introducing the reduction, we first present the polynomial $Q$ in the statement of Theorem 4.3. The original construction in Fournier et al. [11] is for the general class, but over here, we only present a specific case with $r = 2$, which is sufficient for our purpose.

**The Construction of the Polynomial** $Q$     We denote the polynomial $Q$ by $H^{(k,2)}$, which is defined over $2^{2k}$ variables

$$\left\{ x_{\sigma,\tau} : \sigma, \tau \in [2]^k \right\}. \tag{1}$$

The polynomial $H^{(k,2)}$ is recursively defined over intermediate polynomials $H_{u,v}$ for all $(u, v) \in [2]^{\leq k} \times [2]^{\leq k}$ and $|u| = |v|$. Specifically, if $|u| = |v| = k$, then $H_{u,v} = x_{u,v}$; otherwise, $H_{u,v} = \sum_{a=1}^{r} H_{u1,va} H_{u2,va}$. The final polynomial $H^{(k,2)}$ is defined to be $H_{\emptyset,\emptyset}$. Observe that the degree of $H^{(k,2)}$ is $2^k$, and it contains $2^{2^k-1}$ monomials.

Given a minimum tree-structured PC $\Pi$, which computes $P$ and is of depth $o(k)$, we remove all of its leaves that represent negation variables and call this pruned network $\Pi'$; without leaves representing negation variables, $\Pi'$ is just a standard arithmetic formula. Clearly, $|\Pi'| \leq |\Pi|$, and the next proposition reveals the polynomial computed by $\Pi'$, and its proof is in Appendix B.

**Proposition 4.4.** *The arithmetic formula $\Pi'$ computes $H^{(k,2)}$.*

Having all the necessary ingredients, we are now ready to conclude this section by proving Theorem 4.1, the main result of this section.

*Proof of Theorem 4.1.* The proof of Theorem 4.3 in Fournier et al. [11] uses the polynomial class $H^{(k,r)}$ as the hard polynomial $Q$ in the statement, in particular, with $r = 2$, $n = 2^{2k}$ and $d(n) =$

$\sqrt{n} = 2^k$. Note that the depth of $\Pi'$ is $o(\log d) = o(k)$, and the degree of $H^{(k,2)}$ is $d = 2^k$, so the conditions in the statement of Theorem 4.1 are indeed satisfied. Since $\Pi'$ is obtained from $\Pi$ by removing leaves, we obtain the following inequality that concludes the proof:

$$|\Pi| \geq |\Pi'| \geq n^{\omega(1)}. \qquad \blacksquare$$

## 5    Conclusion

In this paper we have shown that given a network polynomial with $n$ variables that can be efficiently computed by a DAG-structured PC, we can construct a tree PC with at most quasi-polynomial size and is no deeper than $O(\log n)$. On the flip side, we have also shown that there indeed exists a polynomial that can be efficiently computed by a $\text{poly}(n)$-sized PC without a depth restriction, but there is a super-polynomial separation if we restrict the depth of the tree to be $o(\log n)$. Our results make an important step towards understanding the expressive power of tree-structured PCs and show that a quasi-polynomial upper bound is possible. However, the lower bound is still largely open, and we have only shown a separation under a specific depth restriction. One potential direction for the future work are discussed below: although the upper bound $n^{O(\log n)}$ is quasi-polynomial, it is still prohibitively large as $n$ grows. The construction outputs a tree of depth $O(\log n)$, which would be considered as a shallow tree. Is it possible to further reduce the size of the tree, possibly in the cost of a larger depth?

## Acknowledgement

HZ would like to thank the support from a Google Research Scholar Award.

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

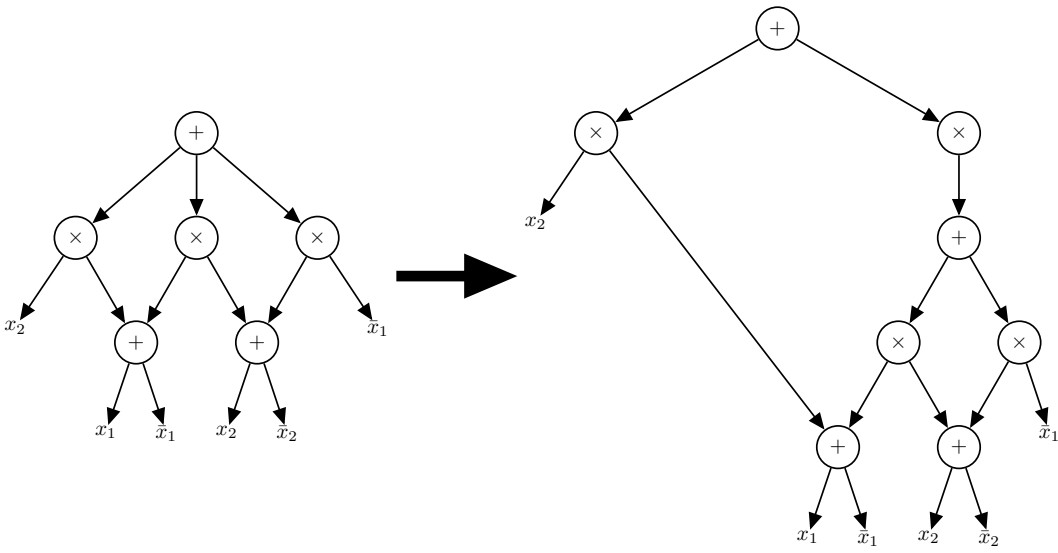

Figure 2: The process of transforming a non-binary DAG-structured PC to a binary one that computes the identical network polynomial. We omit the edge weights for simplicity.

## A  Missing proofs in Section 3

In this section we provide the proofs of the lemmas and theorems that are not included in the main text. For better readability, we first restate the statements and then provide the proofs.

### A.1  Proof of Lemma 3.2

Given a depth-$D$ network with $V$ nodes and $E$ edges, we scan over all its nodes. If a sum node has more than two children, say $M_1, \cdots, M_k$, then keep $M_1$ and create a product node, whose only child is an intermediate sum node. The intermediate sum node has two children: $M_2$ and another just created intermediate sum node. Keep going and until an intermediate sum node has $M_k$ as the only child.

The operation is the same if a product node has more than two children by just exchanging sum and product. Note that for one operation for a node with $k$ children, the depth increases by $2(k-1)$, and $2(k-1)$ nodes and edges are added. Once we do the same for all nodes, the number of increased depth, nodes, and edges are upper bounded by

$$2 \times \left( \sum_{N \in V} \text{out-degree of node } N \text{ if } N \text{ has more than two children} \right) - 2V \leq 2E - 2V \in O(E).$$

In fact, for depth, this upper bound is very conservative because, for example, if a node has three children, one of its children again has three children. After we operate on both of them, the depth increases by four only. A better upper bound is $O(M) \leq O(V)$, where $M$ is the maximum out-degree in the original network. It is easy to check that each child of the root computes the same polynomial as before, and so does the new network. Clearly, the new network is still decomposable and smooth if the original network is.

### A.2  Proof of Lemma 3.3

**Lemma 3.3.** *Given a PC $\Phi$, if $v$ and $w$ are two nodes in $\Phi$ such that $\partial_w f_v \neq 0$, then $\partial_w f_v$ is a homogeneous polynomial over the set of variables $X_v \setminus X_w$ of degree $\deg(v) - \deg(w)$.*

*Proof.* Clearly, $\partial_w f_v \neq 0$ implies that $w$ is a descendant of $v$. We prove the statement by induction on $L$, the length of the longest directed path from $v$ to $w$. If $L = 0$, i.e. $w = v$, then $\partial_w f_v = 1$ and the statement trivially holds. Suppose the statement is true for all $L$ and now the longest distance from $v$ to $w$ is $L + 1$. We prove the statement by discussing two cases, whether $w$ is a sum or product node.

**Case I: $w$ is a sum node.** We first assume $w$ is a sum node, and its parent inside this particular path $v \rightsquigarrow w$ is $u$, whose children are $w$ and $w'$. We write $\overline{f}_v$ as the polynomial if we substitute $w$ with $y$, and $\widehat{f}_v$ as the polynomial if we substitute $u$ with $y$. Note that if we write them as functions with respect to $y$, then $\overline{f}_v(y) = \widehat{f}_v(y \cdot f_{w'})$, and hence

$$\partial_w f_v = \frac{\partial \overline{f}_v(y)}{\partial y} = \frac{\partial \widehat{f}_v(y \cdot f_{w'})}{\partial y} = \frac{\partial \widehat{f}_v(y \cdot f_{w'})}{\partial (y \cdot f_{w'})} \cdot f_{w'} = \partial_u f_v \cdot f_{w'}. \tag{2}$$

By the inductive hypothesis, $\partial_u f_v$ is a homogeneous polynomial over the set of variables $X_v \setminus X_u$ of total degree $\deg(v) - \deg(u)$, so $\partial_w f_v$ must also be homogeneous, and its degree is $\deg(\partial_u f_v) + \deg(w') = \deg(v) - \deg(u) + \deg(w') = \deg(v) - \deg(w) - \deg(w') + \deg(w') = \deg(v) - \deg(w)$, and it is over variables $(X_v \setminus X_u) \cup X_{w'} = (X_v \setminus (X_w \sqcup X_{w'})) \cup X_{w'} = X_v \setminus X_w$.

**Case II: $w$ is a product node.** Next, assume $w$ is a product node. In this case, $u$ is a sum node and $\deg(u) = \deg(w) = \deg(w')$, and $X_u = X_w = X_{w'}$. Let the weight of the edge $u \to w$ be $a$, and the weight for $u \to w'$ be $b$. Then, $\overline{f}_v(y) = \widehat{f}_v(ay + bf_{w'})$, and

$$\partial_w f_v = \frac{\partial \overline{f}_v(y)}{\partial y} = \frac{\partial \widehat{f}_v(ay + bf_{w'})}{\partial y} = a \cdot \frac{\partial \widehat{f}_v(ay + bf_{w'})}{\partial (ay + bf_{w'}))} = a \cdot \partial_u f_v. \tag{3}$$

Clearly, by the inductive hypothesis, both $\partial_u f_v$ and $\partial_w f_v$ are homogeneous, and they have the same degree and set of variables. Specifically, $\deg(\partial_w f_v) = \deg(\partial_u f_v) = \deg(v) - \deg(u) = \deg(v) - \deg(w)$, and $X_{w,v} = X_{u,v} = X_v \setminus X_u = X_v \setminus X_w$. ∎

### A.3 Proof of Lemma 3.4

**Lemma 3.4.** *Let $v$ be a product node and $w$ be any other node in a PC $\Phi$, and $\deg(v) < 2\deg(w)$. The children of $v$ are $v_1$ and $v_2$ such that $\deg(v_1) \geq \deg(v_2)$. Then $\partial_w f_v = f_{v_2} \cdot \partial_w f_{v_1}$.*

*Proof.* Clearly, $\deg(v) = \deg(v_1) + \deg(v_2)$. Therefore, since $\deg(v) < 2\deg(w)$, we have $\deg(v_2) < \deg(w)$; by Lemma 3.3, we have $\partial_w f_{v_2} = 0$, and the conclusion follows directly because of the chain rule. ∎

### A.4 Proof of Lemma 3.6

First, observe that with such choice of $m$, we have $\mathbf{G}_m \cap \Phi_v \neq \emptyset$. Write $v_1$ and $v_2$ as the children of $v$. If $\deg(v_1) \leq m$ and $\deg(v_2) \leq m$, then $v \in \mathbf{G}_m$. Otherwise, assume without loss of generality that $\deg(v_1) \geq \deg(v_2)$ and $\deg(v_1) > m$. Keep reducing and there will be a position such that the condition of being a member in $\mathbf{G}_m$ holds.

We now prove the statement by induction on $L$, the length of the longest directed path from $v$ to $\mathbf{G}_m$, i.e. $L = \max_{v' \in \mathbf{G}_m} \text{dist}(v, v')$. If $L = 0$, then $v \in \mathbf{G}_m$ and all other nodes in $\mathbf{G}_m$ (if any) are not descendants of $v$. Therefore, if $t \in \mathbf{G}_m$ and $t \neq v$, we have $\partial_t f_v = 0$. Clearly, $\partial_v f_v = 1$, so

$$f_v = f_v \cdot \underbrace{\partial_v f_v}_{=1} + \sum_{t \in \mathbf{G}_m : t \neq v} f_t \cdot \underbrace{\partial_t f_v}_{=0} = \sum_{t \in \mathbf{G}_m} f_t \cdot \partial_t f_v. \tag{4}$$

Now suppose the statement is true for all $L$, and now the longest directed path from $v$ to $\mathbf{G}_m$ has length $L + 1$.

**Case I: $v$ is a sum node.** First, assume $v$ is a sum node and $f_v = a_1 f_{v_1} + a_2 f_{v_2}$. Recall that, since $v$ is a sum node, we have $m < \deg(v_1) = \deg(v_2) = \deg(v) \leq 2m$, so we may apply the inductive hypothesis on $v_1$ and $v_2$. Therefore,

$$f_{v_1} = \sum_{t \in \mathbf{G}_m} f_t \cdot \partial_t f_{v_1}; \quad f_{v_2} = \sum_{t \in \mathbf{G}_m} f_t \cdot \partial_t f_{v_2}. \tag{5}$$

Hence, using the chain rule of the partial derivative, we have

$$f_v = a_1 f_{v_1} + a_2 f_{v_2} = \sum_{t \in \mathbf{G}_m} a_1 \cdot f_t \cdot \partial_t f_{v_1} + \sum_{t \in \mathbf{G}_m} a_2 \cdot f_t \cdot \partial_t f_{v_2} \tag{6}$$

$$= \sum_{t \in \mathbf{G}_m} f_t \cdot (a_1 \cdot \partial_t f_{v_1} + a_2 \cdot \partial_t f_{v_2}) = \sum_{t \in \mathbf{G}_m} f_t \cdot \partial_t f_v. \tag{7}$$

**Case II: $v$ is a product node.** Next, assume $v$ is a product node and $\deg(v_1) \geq \deg(v_2)$. If $v \in \mathbf{G}_m$, then the statement trivially holds like the base case, so we assume $v \notin \mathbf{G}_m$, and therefore $m < \deg(v_1) \leq 2m$ and the longest directed path from $v_1$ to $\mathbf{G}_m$ has length $L$, while such a path does not exist from $v_2$ to $\mathbf{G}_m$. So, by the inductive hypothesis,

$$f_{v_1} = \sum_{t \in \mathbf{G}_m} f_t \cdot \partial_t f_{v_1}. \tag{8}$$

By definition, if $t \in \mathbf{G}_m$, then we must have $2 \deg(t) > 2m \geq \deg(v)$, and by Lemma 3.4,

$$f_v = f_{v_1} \cdot f_{v_2} = \sum_{t \in \mathbf{G}_m} f_t \cdot (f_{v_2} \cdot \partial_t f_{v_1}) = \sum_{t \in \mathbf{G}_m} f_t \cdot \partial_t f_v. \tag{9}$$

## A.5 Proof of Lemma 3.7

We again write $v_1$ and $v_2$ as the children of $v$, and again induct on $L$, the length of the longest directed path from $v$ to $\mathbf{G}_m$ in the network. If $L = 0$, then $v \in \mathbf{G}_m$, and same as the previous case, every other node $t$ in $\mathbf{G}_m$ is not a descendant of $v$, which implies $\partial_t f_v = 0$. So,

$$\partial_w f_v = \partial_w f_v \cdot \underbrace{\partial_v f_v}_{=1} + \sum_{t \in \mathbf{G}_m: t \neq v} \partial_w f_v \cdot \underbrace{\partial_t f_v}_{=0} = \sum_{t \in \mathbf{G}_m} \partial_w f_t \cdot \partial_t f_v. \tag{10}$$

Suppose the statement is true for all $L$, and now the longest directed path from $v$ to $\mathbf{G}_m$ has length $L + 1$.

**Case I: $v$ is a sum node.** First, assume $v$ is a sum node and $f_v = a_1 f_{v_1} + a_2 f_{v_2}$. Again, since $v$ is a sum node we may apply the inductive hypothesis on $v_1$ and $v_2$:

$$\partial_w f_{v_1} = \sum_{t \in \mathbf{G}_m} \partial_w f_t \cdot \partial_t f_{v_1}; \quad \partial_w f_{v_2} = \sum_{t \in \mathbf{G}_m} \partial_w f_t \cdot \partial_t f_{v_2}. \tag{11}$$

Again, by the chain rule, we have

$$\partial_w f_v = a_1 \partial_w f_{v_1} + a_2 \partial_w f_{v_2} = \sum_{t \in \mathbf{G}_m} \partial_w f_t \cdot (a_1 \partial_t f_{v_1} + a_2 \partial_t f_{v_2}) = \sum_{t \in \mathbf{G}_m} \partial_w f_t \cdot \partial_t f_v. \tag{12}$$

**Case II: $v$ is a product node.** Now assume $v$ is a product node and $\deg(v_1) \geq \deg(v_2)$. If $v \in \mathbf{G}_m$, then the statement trivially holds like the base case, so we assume $v \notin \mathbf{G}_m$, and therefore $m < \deg(v_1) < 2 \deg(w)$ and the longest directed path from $v_1$ to $\mathbf{G}_m$ has length $L$, while such a path does not exist from $v_2$ to $\mathbf{G}_m$. So, by the inductive hypothesis,

$$\partial_w f_{v_1} = \sum_{t \in \mathbf{G}_m} \partial_w f_t \cdot \partial_t f_{v_1}. \tag{13}$$

Since $\deg(v) < 2 \deg(w)$, and for all nodes $t \in \mathbf{G}_m$, we have $2 \deg(t) > 2m > \deg(v)$, so by applying Lemma 3.4 twice, we have

$$\partial_w f_v = f_{v_2} \cdot \partial_w f_{v_1} = \sum_{t \in \mathbf{G}_m} \partial_w f_t \cdot (f_{v_2} \cdot \partial_t f_{v_1}) = \sum_{t \in \mathbf{G}_m} \partial_w f_t \cdot \partial_t f_v. \tag{14}$$

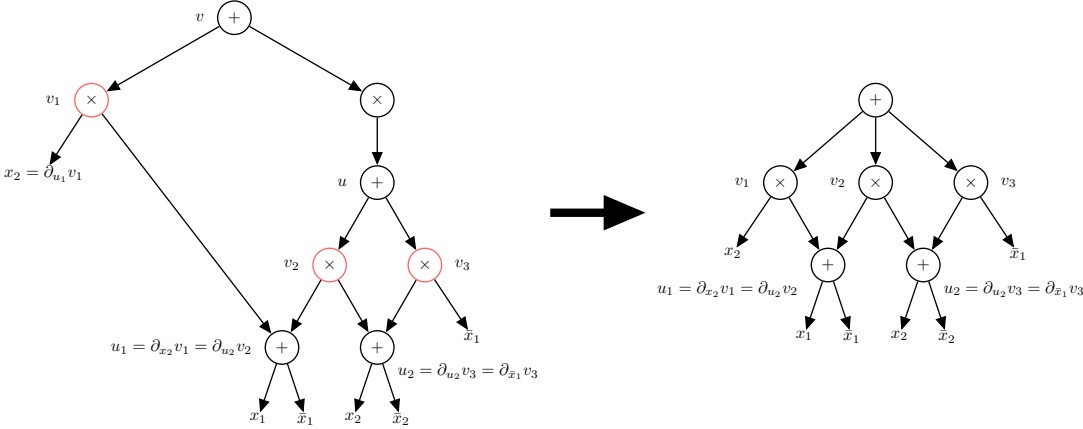

Figure 3: The process of converting an arbitrary DAG to a DAG with depth restriction. The red nodes are those in $\mathbf{G}_2$ and their relationships imply the computational procedure.

### A.6 Proof of Lemma 3.8

Suppose the network is smooth. Recall that if the root of a probabilistic circuit contains $n$ variables, then the network computes a multi-linear polynomial of degree $n$. If the root is a sum node, then its children must be homogeneous with degree $n$. If the root is a product node, then its children must also be homogeneous, otherwise the product node will not be homogeneous.

Conversely, suppose such network is homogeneous. We prove by induction on the depth $d$ of the network. If $d = 1$ and the root is a sum node, then the polynomial must be linear and therefore there can only be one variable $x$ and $\bar{x}$; as a result, this simple network is smooth. Now suppose the statement is true for any $d$, and we have a probabilistic circuit with depth $d+1$. If the root is a product node, we are done because if any sum node had two children with different scopes, the inductive hypothesis would be violated. If the root is a sum node, then every sum node other than the root cannot have two children with different scopes, because each sum node is in the induced sub-network rooted at a grandchild of the root of depth at most $d-1$ so the inductive hypothesis must hold. So, we only need to show $X_R = X_{R_1} = \cdots = X_{R_k}$, where $R_1, \ldots, R_k$ are children of $R$. Since the sub-networks rooted at $R_1, \cdots, R_k$ are decomposable and homogeneous, those sub-networks must be smooth by the inductive hypothesis. Hence, each $R_i$ computes a polynomial of degree $|X_{R_i}|$. If $|X_{R_i}| < n$, then the polynomial computed by $R$ is not homogeneous of degree $n$ and we obtain a contradiction. Therefore, each $R_i$ must contain all of $n$ variables to compute a polynomial of degree $n$ and as a result, we prove $X_R = X_{R_1} = \cdots = X_{R_k}$ and the smoothness of the entire network.

### A.7 Construction of $\Psi$

#### A.7.1 Step one: computing $f_v$ for eligible nodes

During iteration $i + 1$, a polynomial $f_v$ is in consideration if and only if $2^i < \deg(v) \leq 2^{i+1}$. Naturally we shall apply Lemma 3.6, and therefore choosing an appropriate $m$ and the corresponding $\mathbf{G}_m$ is essential. Here we choose $m = 2^i$. Moreover, we define a set $T = \mathbf{G}_m \cap \Phi_v$ for each $v$ being considered; for every $t \in T$, we use $t_1$ and $t_2$ to denote its children. By Lemma 3.6 and the definition that all nodes in $\mathbf{G}_m$ are product nodes, we have

$$f_v = \sum_{t \in T} f_t \cdot \partial_t f_v = \sum_{t \in T} f_{t_1} \cdot f_{t_2} \cdot \partial_t f_v. \tag{15}$$

Since $t \in T$, we must have $\max \{\deg(t_1), \deg(t_2)\} \leq m = 2^i$, and therefore

$$2^i = m < \deg(t) = \deg(t_1) + \deg(t_2) \leq 2m = 2^{i+1}. \tag{16}$$

Therefore, $\deg(v) - \deg(t) < 2^{i+1} - 2^i = 2^i$ and $\deg(v) < 2^i + \deg(t) < 2\deg(t)$. Hence, $f_{t_1}$, $f_{t_2}$ and $\partial_t f_v$ have all been computed already during earlier iterations. If $\deg(v) = \deg(t)$, then $t$ is a

child of $v$ and $\partial_t f_v$ is the weight of the edge $v \to t$. Therefore, to compute such a $f_v$, we need to add $|T|$ product nodes and one sum node, whose children are those $|T|$ product nodes; apparently, the depth increases by two. If a subset of the three terms $\{f_{t_1}, f_{t_2}, \partial_t f_v\}$ is a constant, then their product will be the weight of the edge connecting the product node $f_{t_1} \cdot f_{t_2}$ and the new sum node.

We now verify the validity of this updated circuit. Because $\Phi$ is decomposable and $t$ is a product node, we conclude $X_{t_1} \cap X_{t_2} = \emptyset$ and $X_t = X_{t_1} \sqcup X_{t_2}$. By Lemma 3.3, we have $X_{t,v} = X_v \setminus X_t = X_v \setminus (X_{t_1} \sqcup X_{t_2})$. Therefore, every summand in Equation (15) is a product node whose children are sum nodes with pairwise disjoint scopes, and thus, the updated circuit must be decomposable as well. Also, since $f_v$ is a homogeneous polynomial, so must be every summand for each $t$. As a result, the updated circuit is also homogeneous. Thanks to Lemma 3.8, the updated circuit is valid.

### A.7.2  Step two: computing $\partial_w f_u$ for eligible pairs of nodes

As discussed earlier, during iteration $i + 1$, a pair of nodes $u$ and $w$ are chosen if and only if $2^i < \deg(u) - \deg(w) \le 2^{i+1}$ and $\deg(u) < 2\deg(w)$. In this case, we fix $m = 2^i + \deg(w)$, and define $T = \mathbf{G}_m \cap \Phi_u$. Clearly, $\deg(w) < m < \deg(u) < 2\deg(w)$, so by Lemma 3.7, we have $\partial_w f_u = \sum_{t \in T} \partial_w f_t \cdot \partial_t f_u$. For each $t \in T$, by definition $t$ must be a product node, and since $t \in \Phi_u$, we have $\deg(w) < \deg(t) \le \deg(u) < 2\deg(w)$. Recall that the children of $t$ are denoted by $t_1$ and $t_2$, and we may assume without loss of generality that $\deg(t_1) \ge \deg(t_2)$. Hence, by Lemma 3.4, we have

$$\partial_w f_u = \sum_{t \in T} f_{t_2} \cdot \partial_w f_{t_1} \cdot \partial_t f_u. \tag{17}$$

Furthermore, we may safely assume $\deg(w) \le \deg(t_1)$, otherwise $w$ is not a descendant of $t_1$ nor $t$ and therefore $\partial_w f_{t_1} = \partial_w f_t = 0$. Next, by analyzing their degrees and differences in degrees, we show that for each $t$, the terms $f_{t_2}$, $\partial_w f_{t_1}$, and $\partial_t f_u$ in that summand have all been computed by earlier iterations or the step one during this iteration $i + 1$.

**Term $f_{t_2}$:**  Since

$$\deg(u) \le \deg(w) + 2^{i+1} \le 2^{i+1} + \deg(t_1) = 2^{i+1} + \deg(t) - \deg(t_2), \tag{18}$$

we have

$$\deg(t_2) \le 2^{i+1} + \deg(t) - \deg(u) \le 2^{i+1}. \tag{19}$$

Hence, $f_{t_2}$ has already been computed during the first step of this current iteration or even earlier.

**Term $\partial_w f_{t_1}$:**  Recall that $\deg(t_1) \le m = 2^i + \deg(w)$, so $\deg(t_1) - \deg(w) \le 2^i$. Moreover, $\deg(t_1) \le \deg(t) \le \deg(u) < 2\deg(w)$. Therefore, the pair $(t_1, w)$ satisfies both requirements to be computed during iteration $i$ or earlier.

**Term $\partial_t f_u$:**  Recall that $\deg(t) > m = 2^i + \deg(w)$, so

$$\deg(u) - \deg(t) < \deg(u) - \deg(w) - 2^i \le 2^{i+1} - 2^i = 2^i, \tag{20}$$

where the second inequality follows from $\deg(u) - \deg(w) \le 2^{i+1}$, the requirement of choosing $u$ and $w$ for iteration $i + 1$. Finally,

$$\deg(u) \le 2^{i+1} + \deg(w) < 2 \cdot \underbrace{(2^i + \deg(w))}_{=m < \deg(t)} < 2\deg(t). \tag{21}$$

These two facts together ensure that $\partial_t f_u$ must have been computed during iteration $i$ or earlier.

Finally, we verify the validity of the updated circuit after this step. The new objects introduced in this step are only $|T|$ product nodes whose children are $f_{t_2}$, $\partial_w f_{t_1}$, and $\partial_t f_u$ for each $t \in T$, and one sum node whose children are those $|T|$ product nodes. It is easy to see that the sets $X_{t_2}$, $X_{t_1} \setminus X_w$ and $X_u \setminus X_t$ are pairwise disjoint since $X_w \subseteq X_{t_1}$ and $X_{t_1} \cap X_{t_2} = \emptyset$; therefore, the updated circuit is indeed decomposable. By Lemma 3.3, all three terms in each summand are homogeneous, and therefore the new circuit is also homogeneous, and consequently, it is valid again by Lemma 3.8.

# B  Missing proofs in Section 4

## B.1  Proof of Proposition 4.2

Before writing the rigorous proof, we first fix some terminologies. In this proof, we refer the layer of all indicators constructed in step one as layer zero, and each set of nodes constructed in one of steps three, four, and five as one layer above. A negation indicator added in step six does not belong to any layer. Therefore, when we consider a layer of sum nodes, the negation indicator leaves whose parents are product nodes on the next layer are not in consideration. Step six augments the scope of every product node; for any product node $v$, we use $v'$ to denote the same vertex before being augmented during step six. To prove this statement, we first prove an equivalent condition for $P^*$ to be valid, and then show $P^*$ satisfies the equivalent property.

**Lemma B.1.** *Validity of $P^*$ is equivalent with the following statement:*

*In $P^*$, every product node and its sibling have the same scope. If two product nodes are on the same layer but not siblings, then they have disjoint scopes.*

*Proof.* Suppose the statement holds, then for any sum node, its two children are product nodes and siblings, so they have the same scope; for any product node $v$, denote its children by $w$ and $w'$, and their children by $\{w_1, w_2\}$ and $\{w'_1, w'_2\}$, respectively. Clearly, $w_i$ and $w'_i$ are siblings and have the same scope for any $i \in \{1, 2\}$, but if $j \neq i$, then $w_i$ and $w'_j$ have disjoint scopes. Therefore, $\mathrm{scope}(w) = \mathrm{scope}(w_1) = \mathrm{scope}(w_2)$ and $\mathrm{scope}(w') = \mathrm{scope}(w'_1) = \mathrm{scope}(w'_2)$ are disjoint, as desired.

Conversely, suppose $P^*$ is decomposable and smooth. For any pair of product nodes which are siblings, they share a unique parent and thus have the same scope due to smoothness. Now suppose we have two product nodes $v$ and $w$, which are on the same layer but not siblings. We prove a useful fact: If two product nodes are on the same layer $2j + 1$ for some $1 \leq j \leq k$, then $\deg(v) = \deg(w) = 2^{2j+2}$. When $j = 1$, we know that initially every product node on layer one has two leaf children, so adding two negation indicators enforce that every product node on that layer has degree four. Assume the statement is true for all $j$, and we now consider those product nodes on layer $2(j + 1) - 1 = 2j + 1$. By the inductive hypothesis, every product node on layer $2j - 1$ has degree $2^{2j}$, and therefore every sum node on layer $2j$ also has degree $2^{2j}$. If $u$ is a product node on layer $2j + 1$ with the sibling $u^*$, we have $\deg(u') = \deg((u^*)') = 2^{2j+1}$. Step six ensures that $\deg(u) = \deg(u^*) = \deg(u') + \deg((u^*)') = 2^{2j+2}$.

If they share an ancestor that is a product node, then their scopes are disjoint due to decomposability. On the other hand, suppose their only common ancestor is the root, whose children are denoted by $a_1$ and $a_2$, then without loss of generality, we may assume that $v$ is a descendant of $a_1$ and $w$ is a descendant of $a_2$. Because $P^*$ is valid, it must be homogeneous and we have $\deg(a_1) = \deg(a_2)$. The fact we proved in the previous paragraph implies that $\deg(a'_1) = \deg(a'_2) = 2^{2k-3+2} = 2^{2k-1}$. In other words, step six increases the degree of $a'_1$ and $a'_2$ by $2^{2k-1}$ each. Because the whole tree $P^*$ is decomposable, the increase in $\deg(a'_1)$ is exactly $2^{2k-1} = |\mathrm{scope}(a'_2)|$, and vice versa. Due to smoothness, $\{X_1, \cdots, X_{2^{2k}}\} = \mathrm{scope}(a'_1) \cup \mathrm{scope}(a'_2)$, and thus $\mathrm{scope}(a'_1) \cap \mathrm{scope}(a'_2) = \emptyset$. Finally, since $\mathrm{scope}(v) \subseteq \mathrm{scope}(a'_1)$ and $\mathrm{scope}(w) \subseteq \mathrm{scope}(a'_2)$, we must have $\mathrm{scope}(v) \cap \mathrm{scope}(w) = \emptyset$. ∎

Now we prove Proposition 4.2 by showing that $P^*$ indeed satisfies the equivalent property.

**Proposition 4.2.** *The tree PC $P^*$ is decomposable and smooth.*

*Proof.* Now we prove that $P^*$ satisfies the equivalent statement by induction on the index of the layer containing product nodes only. Using the index above, only the layers with odd indices from $\{2i - 1\}_{i=1}^{k}$ are concerned. For the base case, consider those $2^{2k-1}$ product nodes constructed in step two, denoted by $v_1, \cdots, v_{2^{2k-1}}$. For each $1 \leq j \leq 2^{2k-1}$, if $j$ is odd, then following steps two and six, the children of $v_j$ are $\{x_{2j-1}, x_{2j}, \bar{x}_{2j+1}, \bar{x}_{2j+2}\}$. Its only sibling is $v_{j+1}$, whose children are $\{x_{2j+2}, x_{2j+1}, \bar{x}_{2j}, \bar{x}_{2j-1}\}$. Thus, $\mathrm{scope}(v_j) = \mathrm{scope}(v_{j+1}) = \{X_{2j-1}, X_{2j}, X_{2j+1}, X_{2j+2}\}$. The argument is identical if $j$ is even.

On the other hand, suppose $1 \leq r < s \leq 2^{2k-1}$ and two product nodes $v_r$ and $v_s$ are not siblings, i.e. either $s - r > 1$, or $s - r = 1$ and $r$ is even.

**Case I:** $s - r > 1$. In this case, the set $\text{scope}(v_r)$ is $\{X_{2r-1}, X_{2r}, X_{2r+1}, X_{2r+2}\}$ if $r$ is odd, $\{X_{2r-3}, X_{2r-2}, X_{2r-1}, X_{2r}\}$ if it is even; similarly, the set $\text{scope}(v_s)$ depends on the parity of $s$. If $s - r = 2$, then they have an identical parity. If they are both odd, then $\text{scope}(v_s) = \{X_{2(r+2)-1}, X_{2(r+2)}, X_{2(r+2)+1}, X_{2(r+2)+2}\} = \{X_{2r+3}, X_{2r+4}, X_{2r+5}, X_{2r+6}\}$, and is disjoint with $\text{scope}(v_r)$. The argument is identical if they are both even. If $s - r > 2$, then the largest index among the elements in $\text{scope}(v_r)$ is $2r + 2$, and the smallest index among the elements in $\text{scope}(v_s)$ is $2s - 3 \geq 2(r + 3) - 3 = 2r + 3$; hence, $\text{scope}(v_r) \cap \text{scope}(v_s) = \emptyset$.

**Case II:** $s - r = 1$ **and** $r$ **is even.** In this case, $\text{scope}(v_r) = \{X_{2r-3}, X_{2r-2}, X_{2r-1}, X_{2r}\}$ and $\text{scope}(v_s) = \{X_{2s-1}, X_{2s}, X_{2s+1}, X_{2s+2}\} = \{X_{2r+1}, X_{2r+2}, X_{2r+3}, X_{2r+4}\}$ because $s = r + 1$ is odd. Clearly, $\text{scope}(v_r) \cap \text{scope}(v_s) = \emptyset$.

The argument above proves the base case. Suppose the statement holds until the layer $2i - 1$ for some $i < k$, and we now consider layer $2(i + 1) - 1 = 2i + 1$, which contains $2^{2k-2i-1}$ product nodes, denoted by $v_1, \cdots, v_{2^{2k-2i-1}}$. They must have non-leaf children, and we denote these nodes without their leaf nodes by $v'_1, \cdots, v'_{2^{2k-2i-1}}$. By construction, the layer $2i$ below contains $2^{2k-2i}$ sum nodes, denoted by $w_1, \cdots, w_{2^{2k-2i}}$; and the layer $2i - 1$ contains $2^{2k-2i+1}$ product nodes, denoted by $z_1, \cdots, z_{2^{2k-2i+1}}$. For each $1 \leq r \leq 2^{2k-2i-1}$, the product node $v_r$ has children $w_{2r-1}$ and $w_{2r}$, and is their unique parent. Similarly, $w_{2r}$ has children $z_{4r-1}$ and $z_{4r}$ and is their unique parent; $w_{2r-1}$ has children $z_{4r-3}$ and $z_{4r-2}$, and is their unique parent.

We prove a simple fact that will simplify the induction step. We claim that, given two integers $r, s \in \{1, \cdots, 2^{2k-2i-1}\}$ and $r \neq s$, the scopes $\text{scope}(v'_r)$ and $\text{scope}(v'_s)$ are disjoint. Without loss of generality, we assume $r < s$. By construction, $\text{Ch}(v'_r) = \{w_{2r-1}, w_{2r}\}$ and $\text{Ch}(v'_s) = \{w_{2s-1}, w_{2s}\}$; furthermore, $\text{Ch}(w_{2r-1}) = \{z_{4r-3}, z_{4r-2}\}$, $\text{Ch}(w_{2r}) = \{z_{4r-1}, z_{4r}\}$, $\text{Ch}(w_{2s-1}) = \{z_{4s-3}, z_{4s-2}\}$, $\text{Ch}(w_{2s}) = \{z_{4s-1}, z_{4s}\}$. Observe that, if a pair of product nodes belong to one of the four sets above, then they are siblings and have the same scope; if they belong to distinct sets, then they are not siblings and have disjoint scopes. We know that the scope of a node is the union of the scopes of their children, so the four scopes $\text{scope}(w_{2r-1})$, $\text{scope}(w_{2r})$, $\text{scope}(w_{2s-1})$, and $\text{scope}(w_{2s})$ are pairwise disjoint. As a result, the scopes $\text{scope}(v'_r) = \text{scope}(w_{2r-1}) \sqcup \text{scope}(w_{2r})$ and $\text{scope}(v'_s) = \text{scope}(w_{2s-1}) \sqcup \text{scope}(w_{2s})$ are disjoint.

Now we prove the induction step. In the first case, suppose $v_r$ and $v_{r+1}$ are sibling, i.e. $r$ is odd so $v_{r+1}$ is the only sibling of $v_r$. We have shown that $\text{scope}(v'_r) \cap \text{scope}(v'_{r+1}) = \emptyset$. However, step six enforces that $\text{scope}(v_r) = \text{scope}(v'_r) \sqcup \text{scope}(v'_{r+1}) = \text{scope}(v_{r+1})$, as desired.

Next, suppose $1 \leq r < s \leq 2^{2k-2i-1}$ and $v_r$ and $v_s$ are not siblings. Denote the siblings of $v_r$ and $v_s$ by $v_{r'}$ and $v_{s'}$, respectively; by definition, $r' \in \{r - 1, r + 1\}$ and $s' \in \{s - 1, s + 1\}$, depending on the parity of $r$ and $s$. Clearly, the four nodes $v_r, v_{r'}, v_s, v_{s'}$ are distinct, and consequently the four sets $\text{scope}(v_r)$, $\text{scope}(v_{r'})$, $\text{scope}(v_s)$, and $\text{scope}(v_{s'})$ are pairwise disjoint. Step six enforces that $\text{scope}(v_r) = \text{scope}(v'_r) \sqcup \text{scope}(v'_{r'})$ and $\text{scope}(v_s) = \text{scope}(v'_s) \sqcup \text{scope}(v'_{s'})$, which are disjoint as desired. ∎

## B.2 Proof of Proposition 4.4

We first realize the polynomial $P$ returned by Algorithm 3 without adding those leaves representing negation variables. Recall that layer one contains $2^{2k-1}$ product nodes, and before adding negation variables, the bottom layer (layer zero) contains $2^{2k}$ leaves. If for every odd integer $i \in \{1, 3, 5, \ldots, 2^{2k} - 1\}$, we denote the monomial by $f_{i,i+1} = x_i x_{i+1}$, then without adding negation variables, the polynomial can be constructed by the following recursion with $2k + 1$ steps:

- Construct $2^{2k-1}$ monomials $x_1 x_2, \ldots, x_{2^{2k}-1} x_{2^{2k}}$.

- Sum up $2^{2k-2}$ pairs of consecutive monomials, and return $2^{2k-2}$ polynomials with two monomials $x_1 x_2 + x_3 x_4, \ldots, x_{2^{2k}-3} x_{2^{2k}-2} + x_{2^{2k}-1} x_{2^{2k}}$.

- Multiply $2^{2k-3}$ pairs of consecutive polynomials, and return $2^{2k-3}$ polynomials.

- Repeat the operation until only one polynomial is returned.

Observe that this polynomial is exactly $H^{(k,2)}$ defined in Section 4 with an alternative set of indices for the variables ($[2]^k \times [2]^k$ versus $[2^{2k}]$). To prove Proposition 4.4, it is sufficient to show that for

every minimum tree-structured probabilistic circuit $\Pi$ of depth $o(k)$ that computes $P$, the removal of those leaves representing negation variables returns an arithmetic formula that has the same depth and computes $H^{(k,2)}$. To show this, we need the following lemma.

**Lemma B.2.** *In any tree-structured probabilistic circuit $\Pi$ that computes $P$, no sum node has a negation indicator as a child, and no product node has only negation indicators as its children.*

The proof of Lemma B.2 relies on the following lemma on monotone arithmetic formulas.

**Lemma B.3.** *A monotone arithmetic formula computing a homogeneous polynomial must be homogeneous.*

*Proof.* If a formula is rooted at a sum node, then clearly every child of its must be homogeneous. If the root is a product node with $k$ children, then denote the polynomials computed by them as $f_1, \cdots, f_k$ and write $f = \prod_{i=1}^{k} f_i$. Furthermore, for each $i \in [k]$, further assume $f_i$ contains $q_i$ monomials, and write it as

$$f_i = f_{i,1} + \cdots + f_{i,q_i}. \tag{22}$$

Without loss of generality, assume $f_1$ is not homogeneous and $\deg(f_{1,1}) \neq \deg(f_{1,2})$. Then at least two of the monomials of $f$, namely $f_{1,1} \times \prod_{j=2}^{k} f_{j,1}$ and $f_{1,2} \times \prod_{j=2}^{k} f_{j,1}$, must have distinct degrees and therefore destroy the homogeneity of the root. ∎

*Proof of Lemma B.2.* First, observe that in a PC, if a sum node has a leaf as a child, then due to smoothness, it can only have two children, which are negation and non-negation indicators for a same variable.

Suppose $\Pi$ does have a sum node $u$ that has a negation indicator $\bar{x}_i$ as a child. Observe that, if we replace all negation indicators with the constant one, then the resulting tree is still monotone and computes $F$, which is a homogeneous polynomial. The replacement will cause that sum node to compute exactly $x_i + 1$, which is not homogeneous.

Similarly, if a product node $v$ has only negation indicators as its children, then the replacements above force $v$ to compute one. Smoothness enforces that its siblings have the same scope as $v$ does, and without loss of generality we may assume none of its siblings computes the same polynomial as $v$ does, so their degrees are higher than one. As a result, the replacements of all negation indicators to one will force the parent of $v$ to compute a non-homogeneous polynomial, which contradicts Lemma B.3. ∎

Now Proposition 4.4 can be confirmed, because the removal strategy will indeed produce a tree that computes $H^{(k,2)}$, and no internal nodes will be affected, because Lemma B.2 ensures that, no internal node in $\Pi'$ computes a constant one and can be removed.

## C Pseudocodes

---

**Algorithm 4:** Construction of $\mathbf{G}$

---

**Data:** a binary DAG-structured PC $\Phi$ with $V$ nodes and $n$ variables

**Result:** For each $i = 1, \cdots, \log n$, a set of nodes $\mathbf{G}_{2^i}$ and for selected $w$, a set of nodes $\mathbf{G}_{2^i,w}$.

$T \leftarrow \emptyset$; $P_i \leftarrow \emptyset, Q_i \leftarrow \emptyset, \mathbf{G}_{2^i} \leftarrow \emptyset, \mathbf{G}_{2^i,w} \leftarrow \emptyset$ for $i \in \{1, \cdots, \log n\}$ and $w \in \Phi$; $\Phi_v \leftarrow \emptyset$ for $v \in \Phi$. Scan all nodes from the bottom and calculate the degree of each $f_v$.

During the scanning, extract all weights of edges from a sum node to a product node.

**for** *nodes $v$ in $\Phi$* **do**

    $\Phi_u \leftarrow \Phi_u \cup \{v\}$ if $u$ is a parent of $v$. **for** $i = 1$ *to* $\log n$ **do**

        **if** $2^i < \deg(v) \le 2^{i+1}$ **then**

            |  $P_i \leftarrow P_i \cup \{v\}$.

        **end**

        **if** *$v$ is a product node **and** $\deg(v) > 2^i$ **and** $\deg(v_1) \le 2^i$ **and** $\deg(v_2) \le 2^i$* **then**

            |  $\mathbf{G}_{2^i} \leftarrow \mathbf{G}_{2^i} \cup \{v\}$.

        **end**

        **for** *other nodes $w$ in $\Phi$* **do**

            **if** $2^i < \deg(v) - \deg(w) \le 2^{i+1}$ *and* $\deg(v) < 2 \deg(w)$ **then**

                |  $Q_i \leftarrow Q_i \cup \{(v,w)\}$

            **end**

            **if** *$v$ is a product node **and** $\deg(v) > 2^i + \deg(w)$ **and** $\deg(v_1) \le 2^i + \deg(w)$ **and** $\deg(v_2) \le 2^i + \deg(w)$* **then**

                |  $\mathbf{G}_{2^i,w} \leftarrow \mathbf{G}_{2^i,w} \cup \{v\}$.

            **end**

        **end**

    **end**

**end**

---

**Algorithm 5:** Construction of the tree

**Data:** a binary DAG-structured PC $\Phi$ with $V$ nodes and $n$ variables

**Result:** a tree-structurd PC with size $2^{O(\log^2 n)}$

$T \leftarrow \emptyset; P_i \leftarrow \emptyset, Q_i \leftarrow \emptyset, \mathbf{G}_{2^i} \leftarrow \emptyset, \mathbf{G}_{2^i,w} \leftarrow \emptyset$ for $i \in \{1, \cdots, \log n\}$ and $w \in \Phi; \Phi_v \leftarrow \emptyset$ for $v \in \Phi; m \in \mathbb{N}$ is not defined yet

Operate Algorithm 4 and return $\mathbf{G}_{2^i}$ and $\mathbf{G}_{2^i,w}$ for all $i \in \{1, \cdots, \log n\}$ and those $w \in \Phi$ that were selected for computing partial derivatives.

**for** $i = 1$ **to** $\log n$ **do**

    $m \leftarrow 2^i$. **for** $v \in P_i$ **do**

        $T \leftarrow \mathbf{G}_{2^i} \cap \Phi_v$;

        **for** $t \in T$ **do**

            | Create a product node $\otimes_t$ computing $f_{t_1} \cdot f_{t_2} \cdot \partial_t f_v$.

        **end**

        Create a sum node $\oplus_v$ that sums over all $\otimes_t$; for $t \in T$ such that $\partial_t f_v$ is a non-zero-or-one constant, the edge $\oplus_v \to \otimes_t$ has weight $\partial_t f_v$.

    **end**

    **for** $(v, w) \in Q_i$ **do**

        $m \leftarrow 2^i + \deg(w), T \leftarrow \mathbf{G}_{2^i,w} \cap \Phi_v$;

        **for** $t \in T$ **do**

            | Create a product node $\otimes_t$ computing $f_{t_2} \cdot \partial_w f_{t_1} \cdot \partial_t f_v$.

        **end**

        Create a sum node $\oplus_{(v,w)}$ that sums over all $\otimes_t$; for $t \in T$ such that $\otimes_t$ contains a constant multiplier, the edge $\oplus_{(v,w)} \to \otimes_t$ has weight of that constant.

    **end**

**end**

Apply the naive duplication to convert the DAG into a tree. Apply Algorithm 2 in [36] to normalize the tree.

