# OpenReview forum: "On the Expressive Power of Tree-Structured Probabilistic Circuits"
_NeurIPS.cc/2024/Conference — NeurIPS 2024 poster_

### Official Review · Reviewer_Tza1 · 2024-07-01

**Soundness:** 3
**Presentation:** 3
**Contribution:** 3
**Rating:** 6
**Confidence:** 3

**Summary:**

The paper studies the how expressive probabilistic circuits (PCs) whose underlying structure is a tree are compared to those whose structure is a DAG. This is motivated by the fact that algorithms for learning PCs usually construct trees, and thus do not potentially take advantage of the potentially more expressive nature of DAGs. The present work shows that DAGs are strictly more powerful in the sense that there are polynomials that can be expressed as DAG-structured PCs of polynomial size but require super-polynomial size as tree PCs. On the other hand, the gap is not exponential, since DAG-structured PCs can be transformed into tree PCs of sub-exponential size, as shown in the paper.

**Strengths:**

The paper is well-motivated and, in general, written well.

PCs are an active area of research.

The results are novel to my knowledge and give theoretical contributions on both lower and upper bounds.

**Weaknesses:**

For the lower bound, depth constraint $o(\log n)$ needs to be imposed on the tree-structure. As such, the lower bound appears to have very few practical implications, since we rarely need the tree PC to be that shallow. Admittedly, the authors briefly discuss this limitation in the paper.

Sometimes, the presentation could be more polished: For example, symbols $v$, $w$, $u$, $v'$, $v_1$, $v_2$, and $t$ are all used for denoting nodes. Could the notation be somehow made more consistent? Some other examples are listed in the next question.

**Questions:**

In Sec. 1.1, you state that "[the] restriction on the graph of using nodes with at most two children ... is not necessary for PCs", suggesting that nodes in PCs can have multiple children in your setting. On the other hand, you define decomposability for product nodes with exactly two children. It would be good to clarify in the paper which of these is the case.

In Sec. 1.2, you briefly mention the notion of expressive efficiency. Perhaps here (or somewhere else), it would be relevant to cite the paper [34] on probabilistic generating circuits (PGCs), since they subsume PCs for Boolean random variables. Also related is their follow-up paper (Broadrick et al., 2024; full reference below), where they show that allowing PCs to have negative weights in sum nodes makes them as expressive as PGCs.

Lemma 3.2 seems like folklore to me, since it is well-known how circuits of arbitrary fan-in can be transformed into, say, fan-in 2.

The linebreak in Lemma 3.6 can be confusing to the reader.

Algorithm 1: Definition of $m_2$ is unclear, since $w$ is not specified.

Oliver Broadrick, Honghua Zhang, Guy Van den Broeck:
Polynomial Semantics of Tractable Probabilistic Circuits. CoRR abs/2402.09085 (2024)

**Limitations:**

Authors discuss the limitations briefly but sufficiently.

---

> ### Author Rebuttal · Authors · 2024-08-04
>
> We are grateful to the reviewer for acknowledging our contributions, and we deeply appreciate for pointing out the cluttered notation and the limitations of the conditional lower bound. The following is our responses for the questions.
>
> > In Sec. 1.1, you state that "[the] restriction on the graph of using nodes with at most two children ... is not necessary for PCs", suggesting that nodes in PCs can have multiple children in your setting. On the other hand, you define decomposability for product nodes with exactly two children. It would be good to clarify in the paper which of these is the case.
>
> In Definition 2.1, the definition decomposability is intended for ANY pair of a product node’s children v_1 and v_2, instead of implying they are the only children of a product node. We will emphasize this in our final draft.
>
> > In Sec. 1.2, you briefly mention the notion of expressive efficiency. Perhaps here (or somewhere else), it would be relevant to cite the paper [34] on probabilistic generating circuits (PGCs), since they subsume PCs for Boolean random variables. Also related is their follow-up paper (Broadrick et al., 2024; full reference below), where they show that allowing PCs to have negative weights in sum nodes makes them as expressive as PGCs.
>
> Thanks for pointing out the references. We will properly acknowledge their contributions in our final draft.
>
> > Lemma 3.2 seems like folklore to me, since it is well-known how circuits of arbitrary fan-in can be transformed into, say, fan-in 2.
>
> We agree. We added this result for the sake of completeness, because our proof for the upper bound result uses previous results in circuit complexity based on circuits with fan-in two. We will add a clarification stating its relation with existing results in our final draft.
>
> > The linebreak in Lemma 3.6 can be confusing to the reader.
>
> Thanks for pointing out this. We will update the typeset to fix this issue.
>
> > Algorithm 1: Definition of $m_2$ is unclear, since $w$ is not specified
>
> Thanks for pointing out this. The definition of $m_2$ is indeed dependent on the choice of a pair $(u, w)$. We will move the line “Fix $m_2$ … ” down to ensure clarity.

---

> > ### Comment · Reviewer_Tza1 · 2024-08-09
> >
> > Thank you for the response. I'll keep my score.

---

### Official Review · Reviewer_x52f · 2024-07-08

**Soundness:** 3
**Presentation:** 3
**Contribution:** 2
**Rating:** 6
**Confidence:** 3

**Summary:**

This paper considers the structure expressive power of Probabilistic Circuits (PCs) from a theoretical perspective. Specifically, this paper explores how PCs with directed acyclic graph (DAG) structures and those with tree-like structures contrast each other in terms of circuit size and expressive power. The contributions include theoretical derivations for an upper bound and a conditional lower bound for DAG-structured and tree-structured PCs. First, the authors show that there exists a sub-exponentially-sized tree-structured PC to represent the same network polynomial as a DAG-structured PC. Second, they show that a tree-structured PC must have a super-polynomial size to represent the same network polynomial as a DAG-structured PC given a restriction on the tree depth.

**Strengths:**

- This paper is well-organized and logically coherent.
- This paper provides novel theoretical insights into a relatively under-explored topic (DAG-structured PCs).
- The theoretical results and algorithmic methods are well-presented.

**Weaknesses:**

- While the results are interesting, I am a bit skeptical about the applicability of this work in structure learning of PCs. In practice, challenges in PC structure learning often arise from issues such as defining appropriate learning objectives, addressing overfitting, and improving the computational efficiency of learning algorithms. The findings of this paper appear to be of little help in addressing these issues.

**Questions:**

- Could the authors provide some new insights on how this work can help further research in PC structure learning?
- When taking into account the computational costs for learning DAG-structured and tree-structured PCs, how does one manage the trade-offs between these costs, the expressiveness of the circuit, and the resulting circuit size?

**Limitations:**

The authors have discussed the limitations of their work.

---

> ### Author Rebuttal · Authors · 2024-08-05
>
> We are sincerely grateful to the reviewer for acknowledging the novelty of our work and its presentation, and we deeply appreciate the insightful questions.
>
> > Could the authors provide some new insights on how this work can help further research in PC structure learning?
>
> We agree with the reviewer that our main focus of this paper is theoretical, but we hope that our results on clarifying the size separation between tree-structured and DAG-structured PCs can inspire future algorithmic innovation on directly learning DAG-structured or deep tree-structured PCs from data. As we state in the second paragraph of Section 1 Introduction, existing research on structure learning algorithms (e.g. references 1, 8, 12, 16, 21, 26) for PCs either only learns shallow tree-structured PCs or DAG-structured PCs by using tree-structured PCs as intermediates. Neither line of work can fully exploit the size advantages of DAG-structured PCs. Hence, from this perspective, our work suggests and encourages future research into designing algorithms to directly learn DAG-structured PCs from data, and this should also help in terms of generalization of PCs, due to the size reduction.
>
> > When taking into account the computational costs for learning DAG-structured and tree-structured PCs, how does one manage the trade-offs between these costs, the expressiveness of the circuit, and the resulting circuit size?
>
> Balancing the trade-offs among those criteria indeed requires careful consideration. In particular, there are two significant points to consider.
> - Tree-structured and DAG-structured PCs are equally expressive, in the sense that there does not exist a distribution that can be represented by one but not the other. Therefore, expressiveness of a circuit’s output usually does not raise concerns.
> - There indeed exists a potential tradeoff between the size of PCs and the complexity of learning algorithms. While to the best of our knowledge there is no existing structure learning algorithm that can directly learn a DAG-structured PC from data, in principle one could control the trade-off based on the underlying complexity of the distribution to be learned.

---

> > ### Comment · Reviewer_x52f · 2024-08-13
> >
> > Thanks for the response. I am happy to keep my score.

---

### Official Review · Reviewer_PAKa · 2024-07-13

**Soundness:** 4
**Presentation:** 3
**Contribution:** 3
**Rating:** 7
**Confidence:** 4

**Summary:**

This paper studies the expressive power (i.e., expressive efficiency) of tree-structured probabilistic circuits (PCs). Specifically, this paper shows that:
- Any decomposable PC over n random variables can be transformed into a tree-structured PC of depth O(log n) with n^{O(log n)} nodes.
- A super-polynomial separation between tree-structured PCs of depth o(log n) and decomposable PCs. Specifically, the authors show the existence of a network polynomial that can be computed by a poly(n)-size decomposable PC but the size of any tree-structured PC of depth o(log n) is lower-bounded by n^{\omega(1)}.

**Strengths:**

- Though the proof of the upper bound relies heavily on the existing depth-reduction algorithm proposed by Valiant et al. [32] and Raz and Yehudayoff [25], it is significant for the study of PCs to show that this depth-reduction algorithm preserves decomposability.

- The lower bound result is already very close to showing an unconditional super-polynomial separation between tree-structured PCs and decomposable PCs and eventually leading to a tight bound.

**Weaknesses:**

n/a

**Questions:**

n/a

---

> ### Author Rebuttal · Authors · 2024-08-04
>
> We are sincerely grateful to the reviewer for appreciating our contributions.

---

### Decision · Program_Chairs · 2024-09-25

**Decision:**

Accept (poster)

**Comment:**

This paper provides nice separation results on the sizes of DAG and tree-structured probabilistic circuits. The reviewers agree that this result may not be immediately useful in learning algorithms, but is nonetheless a valuable theoretical contribution that complements the existing body of work in PCs. All three reviewers advocate for acceptance. I expect that this result will be cited often in future work on PCs, and may lead to even stronger theoretical results, as one reviewer suggests.